# Infective Endocarditis in a Tertiary Hospital in Porto—Is There Anything New?

**DOI:** 10.3390/idr18010004

**Published:** 2025-12-25

**Authors:** Carolina Gomes, Isabel Gomes Abreu, Lurdes Santos

**Affiliations:** 1Faculty of Medicine, University of Porto, 4200-319 Porto, Portugal; isabel.abreu@ulssjoao.min-saude.pt (I.G.A.); maria.lurdes.uci@gmail.com (L.S.); 2Infectious Diseases Department, Centro Hospitalar Universitário São João, 4200-319 Porto, Portugal

**Keywords:** infective endocarditis, epidemiology, prosthetic valve endocarditis, *Enterococcus faecalis*, multidisciplinary care, Portugal

## Abstract

Background/Objectives: Infective endocarditis (IE) remains a severe and complex disease despite advances in diagnosis and treatment. The changing epidemiological profile, with an ageing population, has reshaped its presentation and management. This study describes the epidemiological, clinical and microbiological characteristics of IE at a Portuguese tertiary referral hospital prior to the establishment of a multidisciplinary Endocarditis Team. Methods: A retrospective analysis was conducted including all adult patients diagnosed with definite or possible IE according to the 2015 ESC criteria, admitted to ULS São João, Porto, between January 2019 and December 2023. Data were collected from electronic medical records and included demographic characteristics, comorbidities, microbiology, imaging, surgical indications and outcomes. Results: A total of 143 IE episodes were identified. Median age was 71 years, with a predominance of heterologous material-related infections (81%). *Enterococcus faecalis*, viridans group streptococci and coagulase-negative staphylococci were the most frequent pathogens. Surgical indication was present in 74% of cases, although surgery was not performed in 22% due to comorbidities or frailty, contributing to a high in-hospital mortality rate. Conclusions: This study provides a contemporary overview of IE in Portugal, reflecting an elderly, comorbid population and a predominance of prosthetic disease. The results highlight the need for multidisciplinary management and early surgical decisions, supporting the creation of Endocarditis Teams in tertiary centres.

## 1. Introduction

Infective endocarditis (IE) is a rare but life-threatening disease. Despite improvements in diagnosis and management, contemporary IE increasingly affects elderly patients, with prior cardiac interventions and multiple comorbidities, which may explain the persistently high mortality rate of approximately 25% [1].

In high-income countries, the epidemiology of IE has shifted from rheumatic heart disease to degenerative valve disease, prosthetic material and healthcare-associated infections, largely driven by the increase in life expectancy [2].

Frequent healthcare exposure predisposes patients to nosocomial IE, particularly following invasive procedures such as cardiac catheterizations, CIED implantation or dialysis [3,4,5]. In this scenario, Staphylococcus aureus and Enterococcus species are identified as the predominant pathogens, associated with worse outcomes and higher mortality [3,5,6]. Selected patients may require early surgical intervention [7,8,9], which contributes to the complexity of the management of current IE cases, since many of the surgical candidates are elderly patients and/or with significant comorbidities that can negatively impact the surgical intervention.

Contemporary epidemiological studies on IE in Portugal remain scarce and are mostly retrospective and small-scale [10,11]. Local series from tertiary referral centres are essential to characterize national trends. In our Portuguese context, structural limitations may contribute to poorer outcomes. Unlike in several European countries, there are no nationwide guidelines specifically dedicated to the diagnostic and therapeutic approach to IE, nor are there multidisciplinary endocarditis teams and systematic bacteremia surveillance programmes established across all cardiac reference centres.

Endocarditis Teams have been associated with improved management and outcomes, including reduced mortality, particularly in complex cases [9,12,13,14].

Our aim was to describe the epidemiology and standard of care for patients with IE at Hospital de São João (ULS—São João), one of two referral centres for Cardiac Surgery in the north of Portugal, between 2019 and 2023, including an assessment of the most common causative agents, therapeutic approach and surgical indications, prior to the establishment of a multidisciplinary Endocarditis Team at the same hospital in May 2024.

## 2. Materials and Methods

This retrospective study included all patients with a codified diagnosed episode of IE at Hospital São João between January 2019 and December 2023.

Eligible patients were ≥18 years old, had a documented diagnosis of IE according to ICD-10 and fulfilled the 2015 European Society of Cardiology (ESC) criteria for possible or definite IE. The decision to apply the 2015 and not the 2023 guidelines was a choice related to the fact that the more recent guidelines were published in the end of the last year included in the study, with limited access in our institution and external hospital centres to new imaging diagnosis included in the new guidelines.

Patients that were transferred from other hospital centres and that were only partially treated/observed in Hospital São João were included if sufficient data was available in the clinical records to fulfil the intended analyzed parameters.

Exclusion criteria were IE cases exclusively associated with implanted cardiac devices (pacemakers, ICDs, CRT devices) or left ventricular assist devices (LVADs).

Definitions of infective endocarditis categories followed the 2015 ESC criteria. ‘Definite IE’ and ‘possible IE’ were assigned according to major and minor criteria based on clinical, microbiological and imaging findings. Native valve endocarditis (NVE) was defined as infection involving only native cardiac valves. Due to low numbers, all IE episodes associated with cardiac heterologous material (including prosthetic valves, TAVI and other surgical materials) were grouped as prosthetic valve endocarditis (PVE). Patients with multiple IE episodes during the observation period were included as independent single episodes.

Cases were identified through a systematic search of the institutional electronic medical records (SClínico^®^ (v4, SPMS, Lisboa, Portugal) and JOne^®^ (CHUSJ, Porto, Portugal)) using ICD-10 discharge codes related to infective endocarditis. All retrieved records were then manually reviewed by two authors to confirm that each episode met the 2015 ESC criteria for definite or possible IE.

Data retrieved included age, sex, history of cardiac surgery, IE risk factors, major comorbidities, IE-related complications, presumed infection source, imaging modality establishing diagnosis, involvement of native or prosthetic valves, valvular or prosthetic insufficiency, new or worsening heart failure, causative organism and identification method, antibiotic susceptibility, indication for surgery, reasons for surgery refusal (if applicable), date of surgery, empirical and targeted antibiotic regimens (including timing), use of oral antibiotics, hospital discharge date, and date and cause of death, when applicable. Detailed echocardiographic findings were not systematically collected due to heterogeneity and incomplete availability in the retrospective electronic records, particularly in transferred patients.

Categorical variables are presented in absolute and relative frequency, whereas continuous variables are summarized with median and interquartile range (IQR). All estimates were performed using Microsoft Excel (16.0, Microsoft Corporation, Redmond, WA, USA).

This study was approved by the Ethics Committee for Health of the Centro Hospitalar Universitário de São João and the Faculty of Medicine of the University of Porto

## 3. Results

### 3.1. Overall Results

A total of 143 episodes of IE were included over a 5-year period. The distribution of cases by year is displayed on Figure 1. The demographic characteristics of the study population are summarized in Table 1.

The most frequent risk factor for IE was the presence of heterologous cardiac material, identified in 136 episodes (95% of the episodes included in the study period), followed by previous episodes of IE. Classical risk factors such as rheumatic heart disease were rare, as well as those related with intravenous drug use.

Regarding the most frequently affected valve/prosthetic, the aortic site was the most involved, both in NVE and PVE (37% and 60% of episodes, respectively). The least affected site was, as expected, the pulmonary valve/site, occurring exclusively in patients with previous cardiac surgery/heterologous cardiac material (three episodes, corresponding to 3% of all PVE episodes).

Among episodes of PVE/IE associated with heterologous material, 30 (25.9%) occurred less than 12 months after the surgery/foreign material implantation. The median time between the implantation of heterologous material and the IE episode was 51.0 months (IQR 11.6–105.7), with a maximum of 471 months.

### 3.2. Imaging Diagnosis Findings

Cardiac imaging studies are shown in Table 2.

Among the 113 definite cases, abnormalities were most frequently identified with transesophageal echocardiography (TEE) (39.8%), followed by transthoracic echocardiography (TTE) (24.8%) and combined TTE + TEE (22.1%). Positron emission tomography (PET) scan imaging contributed to diagnosis in 9.7% of cases, and PET combined with TEE in 3.5%. In the 16 possible IE episodes, TEE was again the most frequently abnormal modality (50.0%), followed by TTE (31.3%). PET was rarely used in possible cases, either alone or in combination with TTE (6.3% each).

### 3.3. Microbiological Findings

The causative agent for IE was identified in 127 episodes (88%), corresponding to 102 cases of PVE/heterologous-material associated IE (positive findings in 88% of the total cases in this group) and 24 cases of native-valve IE (88% of the all the episodes of native-valve IE). In six episodes (5%), the infection was polymicrobial, all occurring in patients with prosthetic/cardiac heterologous material.

In 118 cases (93%), the diagnosis was established by identification of the microorganism in blood cultures. In the remaining cases, six had etiological identification only in the surgical removed cardiac valve or prosthesis (three of them by 16S rRNA gene PCR/sequencing and three by conventional microbiological culture); one by serology (a *Coxiella burnetii* IE case); one by microbiological culture in another affected site (bone tissue from spondylodiscitis) and one by 16S rRNA gene PCR/sequencing in whole blood.

In terms of specific etiological agents, Figure 2 outlines the main findings in native valve and prosthetic valve/heterologous material associated IE.

Other agents correspond to *Pseudomonas aeruginosa* (two cases, both in prosthetic related IE); *Coxiella burnetii* (two cases); *Klebsiella pneumoniae* (two cases); *Proteus mirabilis* (two cases); *Escherichia coli* (one episode); *Candida* spp. (three episodes, one episode secondary to *C. albicans*; two episodes secondary to *C. parapsilosis*, all in prosthetic IE); *Morganella morganii* (one episode); *Campylobacter fetus* (one case); *Pasteurella multocida* (one case); *Cutibacterium acnes* (one case); *Kocuria rosea* (one case); *Nocardia nova* (one episode; native-valve IE); *Listeria monocytogenes* (one episode; prosthetic-related IE);

Other *Streptococcus* include *S. agalactiae* (seven episodes); *S. dysgalactiae* (one episode); *S. pneumoniae* (one episode); *S. equi* (one episode).

Enterococcal species identified corresponded to 23 cases of *E. faecalis* IE and 1 case of *E. faecium* EI.

The microbiological etiology of IE episodes in all episodes is showed in Table 3.

In respect to antimicrobial resistance profile, 16 of the 18 episodes (88%) of *Staphylococcus aureus*-associated IE were methicillin susceptible. By contrast, 13 cases (68%) of coagulase-negative Staphylococcus IE cases corresponded to methicillin-resistant isolates. In enterococcal IE episodes, only one isolate was resistant to ampicillin, and it corresponded to the only *E. faecium*, with no identification of vancomycin-resistant strains.

Concerning Gram-negative antimicrobial susceptibilities, in the HACEK-related IE, only one strain was an extended-spectrum beta-lactamases (ESBL) producer. In other Gram-negative-related IE (eleven episodes in total), nine were ESBL negative strains; one was a ESBL positive but carbapenem susceptible *K. pneumoniae* strain and one case corresponded to a *K. pneumoniae* infection with carbapenemase production (KPC sub-type), which occurred in a patient with late (≥12 months) prosthetic IE.

No data collection was possible as to other antimicrobial susceptibilities profile (namely to potentially active oral antimicrobial drugs), due to lack of insufficient retrievable data through the hospital medical record consultation.

### 3.4. Surgical Indications and Overall Results

Surgical indication was present in 106 episodes (74.1%), 87 of them (82%) being in cases of prosthetic/heterologous material-related IE.

The most common indication was uncontrolled infection (*n* = 49; 46%), followed by heart failure (42 episodes; 40%) and embolism prevention in 14 cases (13%). One episode did not specify the indication for surgery. The detailed information can be consulted in Table 4.

In 23 cases (22%), the surgery was not performed. The reasons elicited for not pursuing surgical intervention were major comorbidities in eight cases (35%); poor baseline functional status in eight cases (35%); surgically complex/high surgical risk in five (22%) and death before surgery in one episode (4%). One episode had missing information regarding the reason for refusing surgery.

Concerning the 83 episodes of registered cardiac surgical intervention for IE, the median interval between hospital admission and cardiac surgery for IE was 11 days (IQR 5–25), with the minimum time being 0 days and maximum 65 days (data regarding 81 episodes with sufficient data on surgery times).

### 3.5. Clinical Complications Related to IE and Intra-Hospital Mortality

Complications attributed to IE were defined as major systemic emboli; new onset or worsening heart failure or renal function; osteoarticular infections and mycotic aneurisms. In 88 episodes (61.5%), there was at least one IE-related complication, while 55 episodes (38.5%) had an uncomplicated course.

Acute or decompensated heart failure was the most frequent complication, occurring in 53 episodes (37.1%), which was twice as common in PVE than NVE. Major systemic embolic events or neurologic events (registered cases of ischemic stroke or brain hemorrhage) were described in 43 episodes (30.1%). The median age of affected patients was 71 years (IQR 62.5–78) and 69% of them were male.

Acute kidney injury or decompensated chronic kidney disease occurred in 17 episodes (11.9%) and were referenced usually in the presentation of the disease.

Osteoarticular involvement, in the form of spondylodiscitis, was diagnosed in nine episodes (6.3%), in patients with a median age of 69 years (IQR 65; 83). The majority (four episodes, 44%) occurred in cases of IE by *Staphylococcus aureus*, with documented bacteremia.

Mycotic aneurysms complicated the course of two IE episodes (1.4%): both were cases of prosthetic-associated IE, one of them by *Candida parapsilosis* and the other one by a *viridans* streptococcus.

The median hospital stay, until either discharge or death, was 45 days (IQR 29–66.2).

Overall, there were 50 registered deaths in our study, 12 in patients with IE with indication for medical treatment (24%), 21 in patients with surgical indication and that were operated (42%) and 17 patients (34%) in whom there was a surgical indication, but the procedure was not performed. In half of the cases (25 cases), the cause of death elicited in the medical registry was attributed to IE. Twenty-seven patients died during their hospital stay, corresponding to an intra-hospital mortality rate of 54%.

In the group of patients submitted to surgical cardiac intervention (*n* = 83), 21 (25%) died, 9 (40%) of them during the same hospital episode of the IE episode; in 6 cases, the cause of death was attributed directly to complication of IE and/or cardiac surgery intervention. In two cases, it was unrelated and in one case the information was missing.

Concerning patients that had surgical indication for treatment of IE but did not follow through with surgery due to previously mentioned reasons (*n* = 23), overall mortality was 74% (17 of 23 patients). Seventy-five percent of those deaths occurred during the same hospital episode associated with the IE/diagnosis management. In 10 cases (58%), the cause of death was registered as related directly to IE (5 cases with missing information; 1 case unrelated).

Data on causes and timing of death after hospital discharge were impossible to retrieve through medical records consultation.

## 4. Discussion

Although IE represents an important healthcare challenge, much remains to be understood. This retrospective single centre series provides a contemporary snapshot of IE in a large Portuguese tertiary hospital in ULS-São João, Porto.

Our cohort was characterized by a median age of 71 years and a high prevalence of comorbidities, reflecting the demographic ageing reported across Europe, where the incidence of IE now peaks in patients aged ≥80 years, with a persistent male predominance [6,15]. This contrasts with earlier Portuguese studies, in which the median age was 55.5 ± 12.1 years [10,11].

Classical native predisposing conditions were uncommon, confirming the shift towards device- and procedure-related IE [16,17]. Nearly one quarter of patients had pre-existing heart failure, recognized as the strongest predictor of early mortality [15]. The high comorbidity burden in elderly patients further compounds the risk of adverse outcomes [18,19].

The predominance of PVE (81%) in our series—far exceeding the 20–30% reported in multicenter registries [15] and the 22.6% in Portuguese studies [10,11]—likely reflects referral bias in a tertiary cardiac surgery centre and the inclusion of multiple types of heterologous material within this category. Nonetheless, our study indicates a non-negligible group of elderly patients with significant comorbidities, previously surgically manipulated, in which not only the diagnosis may be more difficult to establish but also in which the decision to operate can be difficult to determine. We believe these results reinforce the need for a multidisciplinary team effort, where the definition of the patient’s functional baseline, surgical risk scores and discussion of imaging findings may help to mitigate delays in appropriate care and surgical intervention.

In our series, most cases fulfilled the criteria for definite IE (83%), while 16% were classified as possible IE according to the modified Duke criteria [20], compatible with European data [15]. Among the definite cases, 94% underwent at least one imaging modality, most frequently TEE.

The predominance of PVE shaped a distinct microbiological profile. Even though the *viridans streptococci* were still the predominant causative agent, *Enterococcus faecalis* was the second most common PVE, reflecting the “enterococcal–staphylococcal” transition described in recent reviews (transition from *S. viridians* formerly related to rheumatic EI, to *E. faecalis* in older patients and, more recently, *S. aureus*) [2,21]. Portuguese studies have reported a similar shift with *S. viridian*s as the most frequent, and *Enterococcus faecalis* emerging in PVE and older patients with increased healthcare exposure [10,11].

Coagulase-negative *staphylococci* were also common, which we relate to the number of IE episodes in our cohort related to prosthetic devices [6,15]. Unfortunately, we could not identify, by means of medical record consultation, the suspected or identified portal of entry for the IE episodes in most cases, limiting the conclusions that could be drafted from these findings.

Within our dataset, IE with no causative identified agent occurred in 12% of cases, in line with the reported 10–30% frequency in contemporary cohorts [15,22,23]. The fact that, in 7% of cases, the diagnosis was only established through culture identification in cardiac excised tissue or alternative foci of infection (spondylodiscitis), as well as through the use of molecular biology techniques such as 16S (in blood and in affected tissue), reinforces the importance of persisting in the adequate retrieval and processing of infected tissues and the growing role of non-conventional microbiological techniques [15,22].

The antimicrobial resistance profile identified in this cohort is concerning. More than half of the coagulase-negative *Staphylococcus* strains were methicillin-resistant, confirming the central role of resistant *staphylococci* in PVE and HCA-IE. *Enterococcal* resistance was rare, with only one ampicillin-resistant *E. faecium* and no vancomycin-resistant strains, although vigilance remains necessary given the global spread of VRE. Gram-negative bacilli were less frequent, and ESBL and carbapenemase-producing strains were rare.

Among the 116 PVE cases, 26% were early PVE (<12 months post-surgery) and 73% were late PVE (>12 months), proportions remarkably similar to those reported in EURO-ENDO (24% and 76%, respectively) [15]. This distinction is clinically meaningful, as early PVE is usually healthcare-related, resulting from perioperative contamination or nosocomial infection, whereas late PVE tends to be community acquired. Early PVE was mainly associated with biofilm-forming organisms such as *Staphylococcus epidermidis*, *Staphylococcus hominis* and *Enterococcus faecalis*, while late PVE displayed a broader microbiological spectrum, including oral streptococci and Gram-negative bacilli.

The median interval between prior cardiac surgery and hospital admission for IE had a median overall interval of 52 months (approximately 4 years), underscoring that the majority of PVE episodes represented late presentations. These findings confirm that the risk of PVE extends many years after valve replacement and highlight the importance of lifelong follow-up [9]. The high number of IE episodes in this group of high-risk patients can justify prospective studies trying to identify preventable risk factors.

In our cohort, IE-related complications occurred in over half of episodes, confirming the ongoing severity of the disease in the contemporary era. Acute or decompensated heart failure was the most frequent complication, followed by embolic and neurological events, mirroring the pattern reported in European studies [6,15], where heart failure remains the leading complication and a key driver for surgical referral.

Surgery remains central to IE management [8]. In our cohort, 74% met at least one Class I indication (heart failure, uncontrolled infection or prevention of embolic events [9]). 78% underwent surgery, compared to 50% reported in other studies, which we also attribute to a probable selection bias [15].

There is a high mortality rate (74%) found in our episodes of IE with surgical indication, but who were not operated. Nevertheless, in our series, surgical refusal was based on the information available on clinical records, with no possible way of being verified. Additionally, we are also aware that there was no systematic use of frailty or functional status assessments tools nor of surgical risk calculators before the implementation of the IE multidisciplinary team in our centre, therefore further limiting the ability to determine whether decisions were concordant with the available guidelines.

Almost all non-operated patients in our cohort died and this pattern was expected. This does not imply that surgery would have altered the outcome in those deemed inoperable, but it highlights the critical need for a more systematic and transparent decision-making process and rigorous definition of which patients are truly unsuitable for surgery, as the refusal is almost invariably associated with a fatal outcome.

A total of 50 patients died during hospitalization, of which 25 (50%) were directly attributed to IE. An overall mortality rate of our studied cases is not possible because we accounted for IE episodes and not patients (since there were patients with more than one IE episode during the observation period). Nonetheless, we can estimate it to be around 35% in the 5-year study period (considering 143 IE episodes identified), likely reflecting older age, prosthetic predominance, but also referral bias. In population-based studies, short-term mortality (30-day) has generally ranged 26–36% [24], which are above those typically reported in multicentre registries [6,15].

This study is limited by its retrospective, single centre design, referral bias, incomplete documentation (detailed information regarding the specific causes of in-hospital deaths was frequently incomplete or inconsistently documented) and lack of post-discharge follow-up (missing data on mortality rate beyond hospital discharge). Exclusion of isolated CIED-related IE and limited use of advanced imaging may also affect generalizability. Empirical antimicrobial therapy lacked detailed documentation, and definitive treatment was not standardized, limiting the use of these data to any pertinent conclusion.

Establishing national and multicentre Portuguese prospective cohort registries should be a priority to track the evolving epidemiology of IE and monitor trends, as well as to explore additional reasons for such high mortality rates and subsequent direct interventions to mitigate this problem. At our institution, the 2024 implementation of a dedicated Endocarditis Team offers an opportunity to assess its impact on decision-making, surgical timing and survival. Simultaneously, surveillance systems should integrate resistance profiles to guide empiric therapy and antimicrobial stewardship. These initiatives are essential to improve patient outcomes and healthcare responsiveness.

## 5. Conclusions

In summary, although our study may not have groundbreaking findings, it provides valuable insight into the evolving landscape of IE in Portugal. We observed a substantial proportion of patients with surgical indication who were not operated, reflecting a frail, elderly population with limited potential benefit from intervention, but very high mortality. The predominance of *Enterococcus* species and coagulase-negative *Staphylococci*, along with the relative scarcity of *S. aureus* and late PVE, points toward a shift in the local epidemiology of IE. These trends highlight the need to better understand the origins of these infections and to reinforce preventive strategies, particularly in vulnerable patients. So, to answer the question, there is indeed something new: a changing face of IE that reflects the realities of modern patient populations.

## Figures and Tables

**Figure 1 idr-18-00004-f001:**
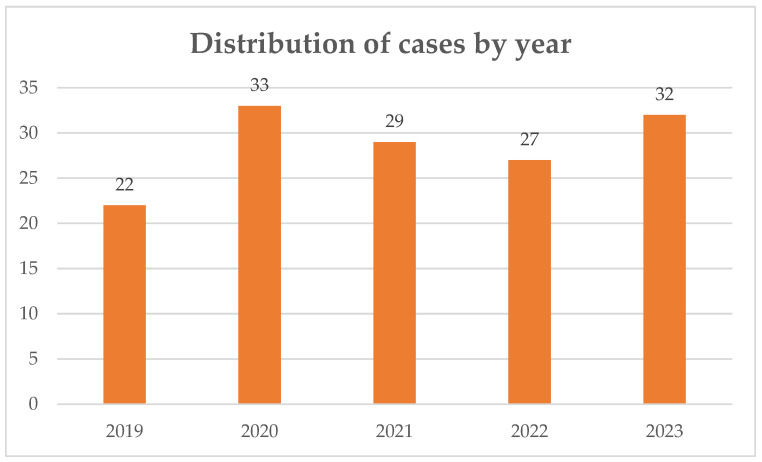
Annual distribution of infective endocarditis episodes recorded at our institution between 2019 and 2023. The y-axis represents the number of episodes per year; x-axis: year.

**Figure 2 idr-18-00004-f002:**
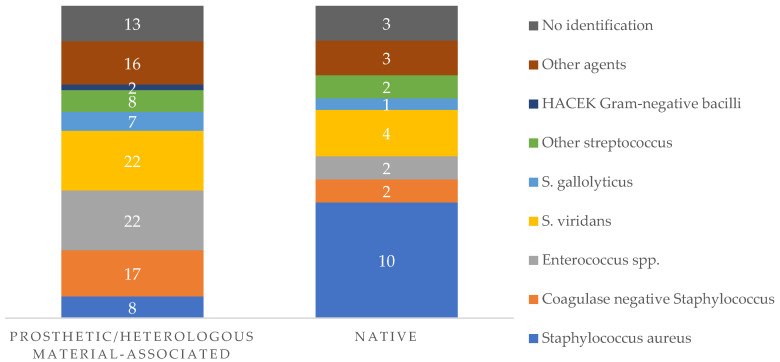
Microbiologic etiology of IE episodes associated with heterologous and native material. Numbers inside the bars represent the absolute number (*n*) of IE episodes attributed to each etiological agent in each group.

**Table 1 idr-18-00004-t001:** Demographic characteristics or comorbidities.

Variable	*n*	% (*n* = 143) ^§^
Demographic characteristics		
Age, median [IQR], years	71 [60.5–76]	---
Male sex	93	65.0
Definite IE	120	83.9
Possible IE	23	16.1
Native-valve IE (NVE)	27	18.9
Prosthetic-valve IE (PVE) *	116	81.1
Risk factors for IE		
Heterologous cardiac material ^Ψ^	136	95.1
≥2 previous interventions	23	16.1
Previous history of IE	12	8.4
Rheumatic heart disease	3	2.1
Non-rheumatic degenerative cardiac valve disease	2	1.4
Use of intravenous drugs (people who inject drugs)	2	1.4
Other ^ϕ^	5	3.5
Co-morbidities	49	34
Chronic heart failure	35	24.5
Chronic kidney disease on haemodialysis	5	3.5
Liver cirrhosis	4	2.8
Solid-organ malignancy on active chemotherapy	2	1.4
HIV infection	2	1.4
Haematological malignancy under active treatment	1	0.7

^§^ Episodes registered in our cohort may correspond to patients with one or more risk factors, so it is expected that in some cases the numbers of risk factors and/or comorbidities exceed the number of IE episodes. * As mentioned in the Methods sections, PVE also includes episodes with simultaneous presence of CIEDs or other heterologous cardiac materials besides cardiac prosthetic valves. ^Ψ^ Each device/foreign material counted as an individual risk factor. ^ϕ^ Other considered risk factors for IE were corrected cardiac congenital disease (*n* = 2); bicuspid aortic valve (*n* = 1); hypertrophic cardiomyopathy (*n* = 1), uncorrected congenital heart disease (*n* = 1).

**Table 2 idr-18-00004-t002:** Diagnostic imaging methods performed in patients with infective endocarditis (N = 143).

Diagnostic Imaging Modality	*n*	% (*n* = 143)
TEE with abnormalities	53	37.1%
TTE with abnormalities	33	23.1%
Combined TTE + TEE	26	18.2%
PET	12	8.4%
PET + TEE	4	2.8%
PET + TTE	1	0.7%
No imaging reported	14	9.8%

**Table 3 idr-18-00004-t003:** Microbiological etiology of IE episodes.

Causative Agent	*n*	% (*n* = 143)
*S. viridans*	26	18%
*Enterococcus*	24	17%
Coagulase negative *Staphylococcus*	19	13%
Other	19	13%
*S. aureus*	18	13%
No identification	17	12%
Other *streptococcus*	10	7%
*S. gallolyticus*	8	6%
HACEK	2	1%

**Table 4 idr-18-00004-t004:** Primary surgical indications (*n* = 106).

Indication	*n*	%
Uncontrolled infection	49	46
Heart failure/valve insufficiency or acute pulmonary edema/cardiogenic shock	42	40
Embolism prevention	14	13
Not documented	1	1

## Data Availability

Data available on reasonable request from the corresponding author; data are not publicly available due to patient privacy.

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
