# Peer review of "Infective Endocarditis in a Tertiary Hospital in Porto—Is There Anything New?"

_2036-7449, 2025, doi:10.3390/idr18010004_

Round 1
Reviewer 1 Report
Comments and Suggestions for Authors
First of all, authors cannot ignore the 2023 ESC Guidelines for IE.
It would be better to focalise on "Definite IE".
Second, many recent articles on the current profile of IE are lacking.
You don't develop the initial work-up.
You don't develop the complications of IE.
Associated malignacies?
Nothing about Emboli?
Nothing about abscesses (spleen, brain).
Are there any relapse?
No data on the causes of in-hospital deaths.
Only 2 infectious aneurysms? What sites? What treatment, what outcome?
What was the time of cardiac surgery from admission. Valve replacement? Bioprosthesis or mechanical prosthesis.
Your experience is very different from that of other tertiary hospitals, especially in Europe.
Regarding the structure of the article:
The introduction is too long . 50 lines should be enough. Many of These data are well known
Reviewer 2 Report
Comments and Suggestions for Authors
Comments and suggestions in the attached file.

Good. The text needs some corrections
Reviewer 3 Report
Comments and Suggestions for Authors
The authors have conducted this epidemiological-based study for Infective endocarditis (IE) at a hospital in Portugal. They have also intended to define the assessment of the most common causative agents, therapeutic approach, and surgical indications, prior to the establishment of a multidisciplinary Endocarditis Team at the same hospital.
Below are some technical queries that need to be addressed:
- The inclusion criteria for the study have to be defined more clearly. For example, the authors have mentioned “Eligible patients were ≥ 18 years old, had a documented diagnosis of IE according to ICD-123 10, and fulfilled the 2015 European Society of Cardiology (ESC) criteria for possible or definite IE. Patients who were transferred from other hospital centers and that were only partially treated/observed in Hospital São João were included if sufficient data was available 126 in the clinical records to fulfill the intended analyzed parameters, under the “materials and methods” section. They need to explicitly define and describe briefly, like how and where (which specialty department database, and what sort of information, etc.) they try to obtain the data, for easy understanding.
- Similarly, adding to the above comment, with regard to data collection and analysis of results, I am wondering if the authors were able to retrieve and collect any information on predisposing cardiac conditions or clinical and echocardiographic reports of the study subjects, along with all the other data that was collected? Since I do not see these relevant criteria listed under the data collection part under the “materials and methods” section.
- Likewise, with regard to the “variables” used for this study, the authors need to clearly define, perhaps as an additional subheading under the “materials and methods” section. For example, how do you define definite IE, possible IE, NVE, etc., like what are signs, symptoms, complications, duration period at the hospital, year of discharge, etc., included for this study?
- Adding to the above comment, did the authors obtain any information or data (co-morbidities) on the subjects’ previous drug history, use of opioid drugs, and other relevant information on any obstructive lung disease, microorganisms, including Streptococci, gram-negative anaerobes, fungus, brucella, etc.
- And were there any IE-compatible complications like ischemic stroke, septic shock, CNS abscess, or meningitis?
- And most importantly, the authors have mentioned “Patients that were transferred from other hospital centers and that were only partially treated/observed in Hospital São João were included if sufficient data was available in the clinical records to fulfill the intended analyzed parameters”. I am wondering if there was any inter- or intra-hospital death data collected?
- Why is the statistical analysis section missing? Details on how variables were represented, measured, and presented should be clearly and explicitly mentioned for clarity. What was the software used for the analysis, what was the regression model used, what is the CI, etc., are all missing.
- Figure 1 is not sufficient. The authors need to label axes properly. And what is the “orange” key series 2?
- And could the authors at least mention the etiological IE agents (microorganisms) in Figure 1?
- Similarly, in Figure 2, the authors may have to use a different template as a bar graph for easy interpretation, and what are the numbers represented inside the bar graph?
- Finally, the authors need to infer the incidence rate and temporal trend of IE in the tertiary hospital in Porto. Perhaps it could be discussed under the discussion section.
- It will be definitely worthwhile if the authors could concentrate on a separate trend graph (survival analysis) on the evolution of annual in-hospital surgical rate and mortality rate (%) among patients hospitalized with IE, in Porto, 2019–2023.
Round 2
Reviewer 1 Report
Comments and Suggestions for Authors
The questions I asked have not been answered.
It is impossible to study the profile of endocarditis without mentioning complications.
Furthermore, the text has not been shortened as requested. The reader is inundated with a series of well-known facts, particularly in the introduction.
Author Response
comment: The questions I asked have not been answered.
It is impossible to study the profile of endocarditis without mentioning complications.
Furthermore, the text has not been shortened as requested. The reader is inundated with a series of well-known facts, particularly in the introduction.We thank the reviewer for this comment and apologize if our previous revision did not sufficiently address the concerns raised.
answer: Thank you for your comment. In the current revised version, we have explicitly included infective endocarditis–related complications in the Results section in line 211, with a dedicated subsection detailing major in-hospital complications, as well as in the Discussion in line 317, where their clinical relevance and impact on outcomes are addressed. This addition allows for a more comprehensive characterization of the endocarditis profile in our cohort.
Furthermore, the manuscript has been substantially shortened, particularly in the Introduction, which was streamlined to remove well-established background information and to focus more clearly on the study rationale and objectives.
Reviewer 2 Report
Comments and Suggestions for Authors
The authors responded coherently and satisfactorily to my comments . Nevertheless I would like to stress my opinion concerning the need for a more-in-depth and refined statistical analysis to improve the reliability of the conclusions provided by the authors.
Author Response
comment: The authors responded coherently and satisfactorily to my comments . Nevertheless I would like to stress my opinion concerning the need for a more-in-depth and refined statistical analysis to improve the reliability of the conclusions provided by the authors.
answer:We thank the reviewer for this positive and constructive comment.
We acknowledge the reviewer’s perspective regarding the potential value of a more in-depth and refined statistical analysis to further strengthen the reliability of the conclusions. However, given the retrospective and descriptive nature of the present study, no additional inferential statistical analyses were planned or performed. The results were therefore intentionally limited to descriptive statistics, aiming to provide an accurate epidemiological and clinical characterization of infective endocarditis in our cohort.
This limitation has been carefully considered and is inherent to the study design. Future studies, ideally multicenter and prospectively designed, will allow for more robust statistical modeling and hypothesis-driven analyses, which may further enhance the strength of the conclusions.
We appreciate the reviewer’s insight, which we believe will be valuable in guiding future research in this field.
Reviewer 3 Report
Comments and Suggestions for Authors
The authors have adequately addressed the comments raised; however, in the future, when reporting or publishing any retrospective patient-epidemiological studies, please try to collect as much data as possible, with a more systematic approach.
Author Response
comment: The authors have adequately addressed the comments raised; however, in the future, when reporting or publishing any retrospective patient-epidemiological studies, please try to collect as much data as possible, with a more systematic approach.
answer: We thank the reviewer for this valuable comment and for the overall positive assessment of our revisions. We fully agree that future retrospective epidemiological studies would benefit from more comprehensive and systematically collected datasets. This recommendation will be carefully considered and incorporated into the design of our future research.